# Effects of Temperature, Scarification, Stratification, Phytohormones, and After-Ripening on the Dormancy and Germination of *Eucommia ulmoides* Oliv. Seeds

**Shiming Deng** [1,2], **Zhijun Deng** [2], **Xiaofeng Wang** [1], **Hai Lu** [1] and **Hua Xue** [1,*]

[1] National Engineering Laboratory for Tree Breeding, College of Biological Sciences and Biotechnology, Beijing Forestry University, Beijing 100083, China; dd7696@163.com (S.D.); wxf801@sina.com (X.W.); luhai1974@163.com (H.L.)

[2] Forestry and Horticulture College, Hubei Minzu University, Enshi 445000, China; dengzhijun@hbmzu.edu.cn

[*] Correspondence: xuehua2013@bjfu.edu.cn

**Abstract:** *Eucommia ulmoides* Oliv., the only member of the family Eucommiaceae, is endemic to China and has great development and utilization prospects. The seeds of *E. ulmoides* show dormancy but the underlying mechanism remains unknown. The aim of this study was to determine the cause of the seed dormancy and provide fundamental knowledge for the breeding, genetic improvement, and conservation of the germplasm resources of this species. According to the seed dormancy classification system developed by Jerry M. Baskin and Carol C. Baskin, we compared the germination percentage between intact seeds and isolated embryos, constructed water absorption curves, and evaluated the germination of seeds treated with scarification, cold/warm-moist stratification, after-ripening during dry storage, and gibberellic acid (GA$_3$). The results showed that the intact seeds germinated only at 10 °C with a low germination percentage of 13.3% whereas the isolated embryos had a high normal germination percentage among a wider range of temperatures. According to the results from the scarified seeds, half seeds, and intact seeds, the seed coat significantly restricted the embryo water absorption. The scarification, after-ripening, cold/warm-moist stratification, and GA$_3$ treatments promoted seed germination. Among them, cold-moist stratification was the most effective method and the temperature range of seed germination increased in both directions from 10 °C with prolonged stratification. The germination percentage increased significantly at constant temperatures with the highest germination percentage of 93.7 $\pm$ 0.3% at 10 °C and a light/dark cycle after 90 days of cold-moist stratification. Therefore, the freshly harvested *E. ulmoides* seeds exhibited a combinational dormancy comprising physical and Type 3 non-deep physiological dormancy, causing limited embryo water absorption by the seed coat and a low embryo growth potential. Given the unique phylogenetic characteristics and utility of *E. ulmoides*, our findings should promote studies of seed dormancy evolution and the development and application of *E. ulmoides* germplasm resources.

**Keywords:** *Eucommia ulmoides*; embryo; dormancy; germination; stratification; after-ripening; phytohormone

## 1. Introduction

Seed dormancy is the inability of viable seeds (or other germination units) to germinate under favorable environmental conditions (e.g., water, temperature, light) during a specific period [1,2]. Seed dormancy evolved in seed plants to regulate the timing of seed germination and seedling establishment, allowing seedlings to avoid unfavorable environmental conditions [3–5].

Seed dormancy is an adaptive trait that gradually developed during plant evolution. Baskin and Baskin added the known dormancy types to the plant phylogenetic tree established by Takhtaian and investigated the evolution of plant seed dormancy [2,5]. There are limited plant species with known seed dormancy types and these species are mostly found

in a handful of genera. The study of seed dormancy in other plant species—especially *E. ulmoides* Oliv., given its phylogenetic significance as the sole species in the family Eucommiaceae—will greatly contribute to the study of the evolution of seed dormancy.

Based on the dormancy classification system of Nikolaeva [6], Baskin and Baskin categorized seed dormancy into five classes: physiological dormancy (PD), morphological dormancy (MD), morphophysiological dormancy (MPD), physical dormancy (PY), and combinational dormancy (PY + PD) according to the causes of seed dormancy [2,5,7]. Each class can be divided into levels and several levels can be further divided into types. PD is the most common dormancy type found in wild plant seeds at present. The main germination barrier is that the growth potential of the embryo is not enough to break through the endosperm, the seed coat, and other surrounding structures. The dormancy time is also the time required by the embryo to accumulate strength or weaken the barrier for germination. Although the embryos of seeds with MD have been differentiated, their development process is incomplete. After the fruits fall off, the embryos in the seeds still need a certain time to continue development, which is the dormancy time. Seeds with MPD have both MD and PD. Seeds with PY are due to the impermeability of the surrounding structure outside the embryo. Seeds with combinational dormancy have both PY and PD [2,5]. The cause of seed dormancy can be determined based on the seed coat permeability and embryo development as well as the responses of seed germination to the phytohormone gibberellic acid ($GA_3$), after-ripening, and stratification [2].

Seed dormancy is a typical quantitative genetic trait that is regulated by many genes and influenced by temperature and other environmental factors [8,9]. Seeds with different dormancy types depend on different methods to release the dormancy, which are generally divided into physical methods, chemical methods, biological methods, and combinational methods. Physical methods mainly include temperature, stratification, and mechanical treatments. In temperature treatments, high, low, or alternative temperature treatments can be tried. For a few seeds, a proper cooling or freezing treatment can promote the degradation of ABA and the synthesis of GA and cytokinin to break the seed dormancy. The response to temperature during seed dormancy and the germination stage varies between species [10–12], which is probably an important cause of plant adaptation to the environment in a long-term evolutionary process. Studies have shown that cold/warm stratification with well-ventilated sand or pumice can promote the dormancy release of almost all kinds of seeds, often more efficient than temperature treatments alone. Stratification is often used as the preferred method to break dormancy especially for embryo-physiological after-ripening seeds. The mechanism of stratification is complex and can promote the morphological development and maturation of embryos, the changing of related hormones in the seeds, the degradation of inhibiting substances, the increase of several enzyme activities, the expression activation of related genes, and the reduction of the sensitivity to ABA in embryos [13,14].

Seed/pericarp scarification is a widely used mechanical method. On the one hand, it increases the water permeability/air permeability of the seed/pericarp to break the physical dormancy; on the other hand, it also releases the mechanical restraint of the seed/pericarp on the embryo germination to break the seed physiological dormancy [15,16]. Chemical methods such as plant hormones, growth regulators, and special chemical reagents can also release the seed dormancy. The hormone regulation theory holds that seed dormancy and germination are mainly regulated by ABA and GA in which ABA induces the dormancy and GA promotes the germination [17,18]. Several plant hormones including GA, ethylene, and cytokinin can break the dormancy for a few species. After-ripening refers to a physiological process in which mature seeds can germinate after they leave the parent plant and undergo a series of physiological and biochemical changes. Dry storage and after-ripening treatments also have good promoting effects on the dormancy release and germination for many plant seeds but the mechanism is still unclear [19,20].

*Eucommia ulmoides* is a deciduous tree species belonging to the family Eucommiaceae and is an endangered relict species endemic to China [21,22]. It is widely distributed in the

subtropical and temperate regions of the southern Hunan, Shaanxi, Sichuan, Chongqing, Henan, and Hubei provinces in China [23]. It is a rare economic tree species with important applications in medicine and industry [24–26]. Seed propagation is primarily used for seedling establishment in large-scale productions but the high level of seed dormancy has seriously restricted the production of *E. ulmoides* [26,27]. Even after a long period of stratification or dry storage, the germination percentage is low and non-uniform growth occurs [28–30], which is ultimately attributed to the limited understanding of the mechanism of seed dormancy. Therefore, a systematic study on the causes of *E. ulmoides* seed dormancy not only has theoretical implications for the further understanding of seed dormancy as an adaptive trait but also has important practical applications.

## 2. Materials and Methods

### 2.1. Seeds

In November 2019, seeds were collected from 20 *E. ulmoides* trees of approximately 30 years of age in a plantation in Yanjiatuo Village, Zhuyang Town, Lingbao City, Henan Province, China (34°27′ E, 110°66′ N; elevation 950 m). Fresh ripe seeds were dried in a room at $20 \pm 2$ °C and 50% relative humidity for 1 w to a moisture content of $0.1 \pm 0.01$ gH$_2$O[g(DW)]$^{-1}$. The remaining seeds were stored at 4 °C.

The seeds of *E. ulmoides* used in this study were samaras with a 1000 grain weight of $85 \pm 2.6$ g, which represented the natural dispersal and germination units of *E. ulmoides*.

### 2.2. Determination of the Moisture Content

The constant temperature oven-drying method recommended by the International Seed Testing Association (ISTA, 2012) was used. Each treatment had four replicates and 50 seeds were used for each replicate. The moisture content was calculated based on the fresh weight.

### 2.3. Determination of the Seed Viability

To evaluate the germination potential of the freshly harvested seeds and the viability of the ungerminated seeds after the germination tests, the viability of the intact seeds was measured using the quick triphenyl tetrazolium chloride (TTC) staining method proposed by Moore [31] Moore (1973). Each treatment had four replicates and 50 seeds were used for each replicate. Embryos that were stained dark red were considered to be viable.

### 2.4. Germination of the Isolated Embryos

The embryos were isolated from the freshly harvested seeds and incubated to determine the dormancy status. Each treatment had four replicates and 25 intact isolated embryos were used in each replicate. The excised embryos were evenly sown in Petri dishes (9 cm in diameter) lined with two layers of filter paper saturated with distilled water and sealed with parafilm to prevent water evaporation. The dishes were incubated at 5, 10, 15, 20, 25, and 30 °C (constant temperature) as well as 12/22, 22/31, 14/22, and 5/10 °C (alternating temperature) with a 12 h light/12 h dark cycle (PPDF = 121 μmol·m$^{-1}$·s$^{-1}$). Light was provided during the warm phase. Embryos with a radicle length $\geq$ 2 mm and green cotyledon were considered to be germinated.

### 2.5. Water Absorption by the Intact Seeds, Scarified Seeds, and Isolated Embryos

To test whether the seed coat and endosperm limited the water absorption, the changes in water absorption by the intact seeds, scarified seeds, and excised embryos were measured and water absorption curves were plotted according to the method of Li [32] Li (2018). Each treatment had four replicates and 30 seeds or isolated embryos were used for each replicate. The seeds or excised embryos were evenly sown in Petri dishes (diameter = 9 mL) lined with two layers of filter paper saturated with distilled water (5 mL) and incubated at 10 °C in the dark. The seeds were weighed at regular intervals that were adjusted according to the progression of the test.

### 2.6. After-Ripening Treatment

Seeds that had been freshly harvested and dried in the shade for 1 week were packed in paper envelopes and placed at 15 °C and 50% relative humidity for dry storage. Samples after 30, 60, 90, and 120 days of dry storage were used for the germination tests. Each treatment had four replicates and 25 seeds were used for each replicate. Freshly harvested seeds dried in the shade for 1 w were used as a control.

### 2.7. Scarification Treatment

Freshly harvested seeds were dried in the shade for 1 week. The seeds were scarified with a scalpel near the radicle or cut transversely into two halves of approximately equal size and the halves with radicles were collected. The germination was tested at an optimum temperature (10 °C) in the dark. Each treatment had four replicates and 25 seeds were used for each replicate. Freshly harvested intact seeds were dried in the shade for 1 week and used as the control.

### 2.8. Stratification Treatment

The freshly harvested seeds were dried in the shade for 1 week, well-combined with perlite (70% moisture content) at a volume ratio of 1:3, sealed in black polyethylene bags, and stored at 5 °C and 15 °C for cold/warm-moist stratification, respectively. Samples were collected after 15, 30, 45, 60, 75, 90, and 120 days of stratification and the germination was tested at 5, 10, 15, 20, 25, 30, and 35 °C in the dark. Each treatment had four replicates and 25 seeds were used for each replicate. Freshly harvested seeds dried in the shade for 1 week were used as the control.

### 2.9. GA$_3$ Treatment

Seeds freshly harvested and dried in the shade for 1 week were soaked for 24 h in GA$_3$ solutions at 0.5, 1, 1.5, 2.0, 2.5, and 3.0 mmol·L$^{-1}$, respectively. The seeds were rinsed three times with distilled water and tested for germination at an optimum temperature (10 °C) in the dark. Each treatment had four replicates and 30 seeds were used for each replicate. Seeds immersed in distilled water for 24 h were used as the control.

### 2.10. Germination Test

Seeds were freshly harvested and dried in the shade for 1 week then treated by scarification, dry storage, stratification, and GA$_3$ and evenly sown in germination boxes containing perlite moistened with distilled water. The seeds were incubated at a constant temperature and/or an alternating temperature in the dark or at a 2 h light/12 h dark cycle (PPDF = 121 μmol·m$^{-1}$·s$^{-1}$). Light was provided during the warm phase. The number of newly germinated seeds was observed and recorded daily. Distilled water was supplemented to maintain the moisture content of the perlite. The germination test was terminated after 30 day and the viability of the ungerminated seeds was examined by TTC staining. Seeds with a 2 mm radicle protrusion were considered to be germinated.

### 2.11. Statistical Analysis

The statistical analysis and plotting were performed using R (4.0) software [33] R (4.0) software. The water absorption of the intact and scarified seeds and the isolated embryos as well as the germination percentages of the seeds treated with scarification, dry storage, stratification, and GA$_3$ were subjected to a one-way ANOVA ($p = 0.05$). Post-hoc multiple comparisons were performed using Student–Newman–Keuls (S–N–K, $p = 0.05$). To ensure the homogeneity of the variance, the seed germination and isolated embryo growth data were arcsine transformed before the statistical analysis. All data were expressed as (mean ± standard error) and rounded to one decimal place.

## 3. Results

### 3.1. Viability and Germination of the Intact Seeds

The intact seeds were freshly harvested and then dried in the shade for 1 week. The seeds germinated only at 10 °C (12 h light/12 h dark) and the germination percentage was 13.3 ± 0.7% (Figure 1, $p < 0.05$) although TTC staining showed a 97.7 ± 0.7% viability of ungerminated seeds (data not shown). The results of the germination experiments showed that the *E. ulmoides* seeds had an obvious dormancy. We then separated the embryos to test whether the isolated embryos were dormant.

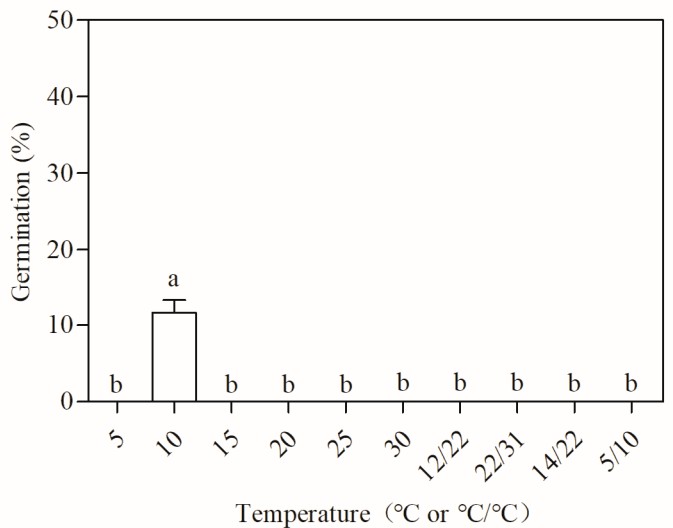

**Figure 1.** Germination of the freshly harvested seeds. Germination was performed at constant temperatures (5, 10, 15, 20, 25, and 30 °C) or alternating temperatures (12/22, 22/31, 14/22, and 5/10 °C) with a 12 h light/12 h dark cycle and PPFD = 121 μmol·m$^{-1}$·s$^{-1}$. Each treatment had four replicates and 30 seeds were used for each replicate. There is no significant difference between the data labeled with the same lower case letters ($p = 0.05$).

### 3.2. Germination of the Isolated Embryos

As shown in Figure 2, the isolated embryos germinated and grew normally under all conditions except 5, 30, and 0/5 °C. High germination percentages (>95%) were found at 10, 15, 20, and 14/22 °C (Figure 2, $p < 0.05$). These results indicated that the isolated embryos were not dormant, suggesting the tissues from the embryos might contribute to the seed dormancy. In subsequent germination tests for the various treatments, 10 °C in the dark was used as the optimum germination condition.

### 3.3. Effect of Scarification on the Dormancy Release and Germination

As shown in Figure 3, half seeds and scarified seeds showed a final germination percentage of 71.7 ± 3.3% and 60.0 ± 7.7%, respectively, significantly higher than the intact seeds (Figure 3, $p < 0.05$). This further confirmed that the coat was a major limiting factor for *E. ulmoides* seed germination.

Seeds with different integrities were tested for germination at 5, 10, 15, 20, 25, and 30 °C as well as 12/22, 22/31, 14/22, and 5/10 °C. The germination percentages of intact seeds, scarified seeds, and half seeds at 10 °C were higher than the other temperatures and the germination percentage of the half seeds was higher than the intact and scarified seeds. A two-way ANOVA showed that both the germination temperature and the coat integrity affected the germination of *E. ulmoides* seeds and these two factors interacted with each other (Table 1, $p < 0.05$).

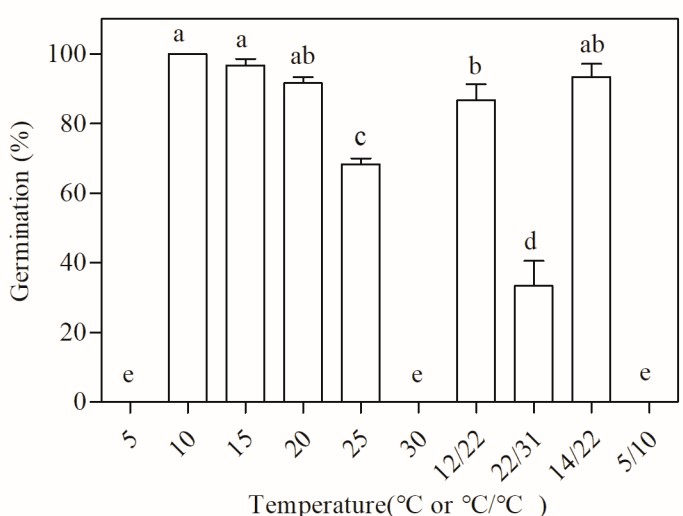

**Figure 2.** Germination of the isolated embryos at constant and alternating temperatures. Germination was performed at constant temperatures (5, 10, 15, 20, 25, and 30 °C) or alternating temperatures (12/22, 22/31, 14/22, and 5/10 °C) with a 12 h light/12 h dark cycle and PPDF = 121 $\mu$mol·m$^{-1}$·s$^{-1}$. Each treatment had four replicates and 30 seeds were used for each replicate. There are no significant differences between the data labeled with the same lower case letters ($p = 0.05$).

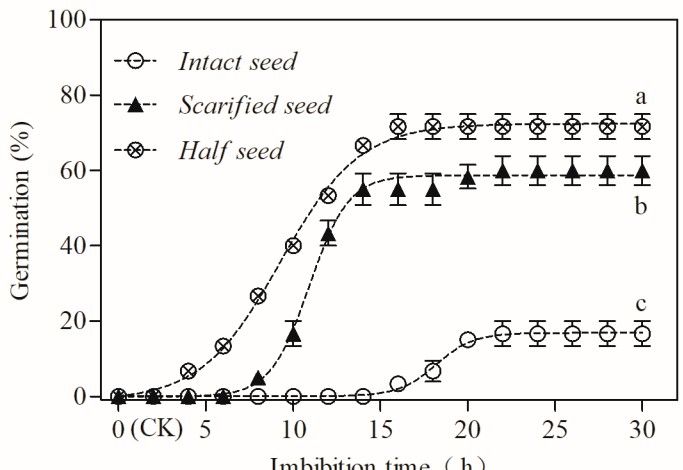

**Figure 3.** Germination process of the scarified seeds, half seeds, and intact seeds. Germination was performed at a constant temperature of 10 °C with a 12 h light/12 h dark cycle and PPDF = 121 $\mu$mol·m$^{-1}$·s$^{-1}$. Each treatment had four replicates and 30 seeds were used for each replicate. There are no significant differences between the data labeled with the same lower case letters ($p = 0.05$).

**Table 1.** Effects of coat integrity and germination temperature and their interaction on the germination percentage.

| Source of Variation | Sum of Squares | D$_f$ | Mean Squares | $f$-Value | $p$-Value |
|---|---|---|---|---|---|
| Seed integrity | 43,575.729 | 3 | 21,787.87 | 992.60 | 0.00 |
| Temperature | 55,389.446 | 9 | 6154.387 | 280.38 | 0.00 |
| Seed integrity × temperature | 22,433.146 | 27 | 1246.29 | 56.78 | 0.00 |

Freshly harvested intact, scarified, and half seeds were incubated at 5, 10, 15, 20, 25, and 30 °C as well as 12/22, 22/31, 14/22, and 5/10 °C for 30 days and the germination percentage was calculated (mean ± standard error). Each treatment had four replicates and 30 seeds were used for each replicate. D$_f$ = degree of freedom ($p = 0.05$).

### 3.4. Water Absorption by the Intact Seeds, Scarified Seeds, and Isolated Embryos

Water availability is the prerequisite for seed germination so the ability to absorb water was measured in the intact seeds, scarified seeds, and isolated embryos. At 4 h, the water absorption of the scarified seeds and isolated embryos was 64.5 ± 1.3% and 45.8 ± 1.8%, respectively, significantly higher than the intact seeds (Figure 4, $p < 0.05$). The water absorption of the scarified seeds did not show an obvious increase (Figure 4, $p > 0.05$) from 4 h on whereas that of the intact seeds and embryos gradually increased from 4 to 36 h. When the water absorption of the intact seeds decreased, the isolated embryos still showed a significant increase in water absorption from 36 to 48 h (Figure 4, $p < 0.05$) and then leveled off. At the end of the experiment, the water absorption of the scarified seeds, isolated embryos, and intact seeds was 72.5 ± 0.5%, 74.0 ± 0.4%, and 62.8 ± 2.3%, respectively. These results suggested that scratching improved the water absorption capacity of the seeds, which may be one of the reasons why it promoted seed germination.

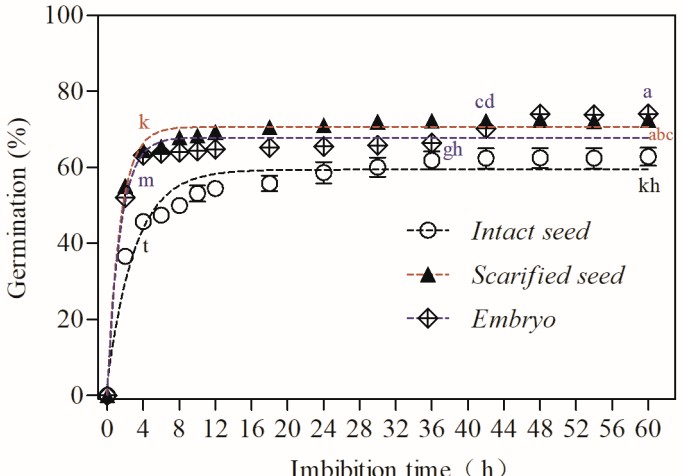

**Figure 4.** Water absorption curves of the intact seeds, scarified seeds, and excised embryos of *Eucommia ulmoides*. The intact seeds, scarified seeds, and excised embryos were allowed to absorb water in the dark at 10 °C, following which they were weighed. Each treatment had four replicates and 30 seeds or excised embryos were used for each replicate. The percentage of water absorption was expressed as mean ± standard error. The 0 in the x-axis denotes the control (CK). There are no significant differences between the data labeled with the same lower case letters ($p = 0.05$).

### 3.5. Effects of Stratification on the Dormancy Release and Germination

According to the data in Table 2, cold-moist stratification at 5 °C was much more effective than warm-moist stratification at 15 °C in promoting the dormancy release and germination ($p < 0.05$). Only a few seeds successfully germinated at 75 days of cold-moist stratification whereas approximately 5% and 11% germinated at 90 and 120 days, respectively (data not shown).

With the extension of cold-moist stratification, the germination percentages for the constant temperatures increased and maximized at 90–120 days of cold-moist stratification. The temperature window of seed germination was broadened with a prolonged cold-moist stratification (Table 2, $p < 0.05$). For example, the temperature range was 10–15 °C at 15 days and 5–25 °C at 75 days. Under the same germination conditions, the seeds treated with warm-moist stratification showed a much lower germination percentage and much narrower germination temperature window when compared with cold-moist stratification (Table 2, $p < 0.05$).

**Table 2.** Germination after stratification.

| Stratification Temperature (°C) | Stratification Duration (Day) | Germination Percentage (%) | | | | | |
|---|---|---|---|---|---|---|---|
| | | 5 °C | 10 °C | 15 °C | 20 °C | 25 °C | 30 °C |
| 5 | 0 | 0.0 [Dc] | 13.3 ± 0.7 [DEa] | 5.0 ± 1.7 [DEFb] | 1.7 ± 1.7 [Dc] | 0.0 [Cc] | 0.0 [Ac] |
| | 15 | 0.0 [Db] | 13.3 ± 0.7 [Ea] | 3.3 ± 3.3 [EFb] | 0.0 [Db] | 0 [Cb] | 0.0 [Ab] |
| | 30 | 0.0 [Dd] | 28.3 ± 6.9 [Da] | 18.3 ± 7.4 [CDab] | 10.0 ± 4.3 [Cbc] | 1.7 ± 1.7 [BCcd] | 0.0 [Ad] |
| | 45 | 0.0 [Dd] | 48.3 ± 9.2 [Ca] | 20 ± 3.9 [Cb] | 10.0 ± 3.3 [Cbc] | 5.0 ± 3.1 [ABcd] | 0.0 [Ad] |
| | 60 | 5.0 ± 1.67 [Cb] | 51.7 ± 5.0 [Ca] | 30.7 ± 5.7 [CEb] | 13.3 ± 2.7 [Cb] | 0.0 [Cc] | 0.0 [Ac] |
| | 75 | 6.6 ± 0.0 [Cc] | 72 ± 4.7 [BCa] | 58.3 ± 6.8 [Ba] | 35.0 ± 3.2 [Bb] | 8.3 ± 1.7 [Ac] | 0.0 [Ad] |
| | 90 | 18.33 ± 1.67 [Bd] | 93.7 ± 0.3 [ABa] | 61.7 ± 6.3 [Bb] | 36.7 ± 3.3 [Bc] | 8.3 ± 1.7 [Ac] | 0.0 [Ae] |
| | 120 | 26.7 ± 1.92 [Ac] | 92.7 ± 0.7 [Aa] | 83.3 ± 7.2 [Aa] | 46.7 ± 6.4 [Ab] | 10.0 ± 1.7 [Ac] | 0.0 [Ad] |
| 15 | 0 | 0.0 [Da] | 0.0 [Fa] | 0.0 [Fa] | 0.0 [Da] | 0.0 [Ca] | 0.0 [Aa] |
| | 15 | 0.0 [Db] | 0.0 [Fb] | 5.7 ± 3.2 [EFa] | 0.0 [Db] | 0.0 [Cb] | 0.0 [Ab] |
| | 30 | 0.0 [Dc] | 20.0 ± 13.6 [DEa] | 6.7 ± 2.7 [DEFa] | 1.7 ± 1.7 [Dab] | 0.0 [Cc] | 0.0 [Ac] |
| | 45 | 0.0 [Db] | 11.7 ± 4.2 [Ea] | 0.0 [Fb] | 0.0 [Db] | 3.3 ± 3.3 [BCb] | 0.0 [Ab] |
| | 60 | 0.0 [Da] | 0.0 [Fa] | 0.0 [Fa] | 0.0 [Da] | 0.0 [Ca] | 0.0 [Aa] |
| | 75 | 0.0 [Db] | 10.0 ± 0.0 [Ea] | 3.3 ± 3.3 [EFb] | 1.7 ± 1.7 [Db] | 1.7 ± 1.7 [BCb] | 0.0 [Ab] |
| | 90 | 0.0 [Da] | 6.7 ± 4.7 [EFa] | 0.0 [Fa] | 0.0 [Da] | 0.0 [Ca] | 0.0 [Aa] |
| | 120 | 0.0 [Da] | 1.7 ± 1.7 [Fa] | 0.0 [Fa] | 0.0 [Da] | 0.0 [Ca] | 0.0 [Aa] |

Seeds were treated with stratification at 5 °C or 15 °C in the dark for 0, 15, 30, 45, 60, 75, 90, and 120 days, respectively. Seeds were then incubated in the dark at set temperatures for 30 days and the germination percentage was calculated (mean ± standard error). Each treatment had four replicates and 30 seeds were used for each replicate. Multiple comparisons between different stratification durations are indicated by upper-case superscript letters and the same letters denote no significant differences. Multiple comparisons between different stratification temperatures are indicated by lower case superscript letters and the same letters denote no significant differences.

The optimum germination temperature of the *E. ulmoides* seeds was 10 °C according to the germination of the seeds treated with cold-moist stratification for 15–120 days. The optimum cold-moist stratification duration was 60–120 days at a 10 °C germination temperature. Seeds treated with cold moisture stratification for 90 d had a maximum germination percentage of 93.7 ± 0.3% at 10 °C (Table 2). A two-way ANOVA showed a significant interaction between the cold-moist stratification duration and the germination temperature (Table 3, $p < 0.05$).

**Table 3.** Effects of cold-moist stratification duration and germination temperature and their interaction on the germination percentage.

| Source of Variation | $D_f$ | Mean Squares | $f$-Value | $p$-Value |
|---|---|---|---|---|
| Cold-moist stratification duration | 9 | 3266.52 | 82.75 | 0.00 |
| Germination temperature | 5 | 10,079.32 | 255.34 | 0.00 |
| Cold-moist stratification duration × germination temperature | 45 | 315.15 | 7.94 | 0.00 |

Freshly harvested intact, scarified, and half seeds were incubated at 5, 10, 15, 20, 25, and 30 °C as well as 12/22, 22/31, 14/22, and 5/10 °C for 30 days and the germination percentage was calculated (mean ± standard error). Each treatment had four replicates and 30 seeds were used for each replicate. $D_f$ = degree of freedom ($p = 0.05$).

### 3.6. Effect of After-Ripening on the Dormancy Release and Germination

After seeds are separated from the mother plants, after-ripening still occurs. We posited whether this might be the cause of the dormancy of the *E. ulmoides* seeds. As shown in Figure 5, after-ripening significantly promoted seed germination (Figure 5, $p < 0.05$). The germination percentage of the freshly harvested seeds was only 13.3 ± 0.7% but reached 25.2 ± 1.3%, 23.3 ± 1.1%, 21.7 ± 1.2%, and 20.0 ± 0.8% after 30, 60, 90, and 120 days of after-ripening, respectively. The germination percentage increased and then decreased; the germination percentage was the highest after 30 days of after-ripening (Figure 5). The results showed that a short period of after-ripening could be an important physiological mechanism of *E. ulmoides* seed dormancy.

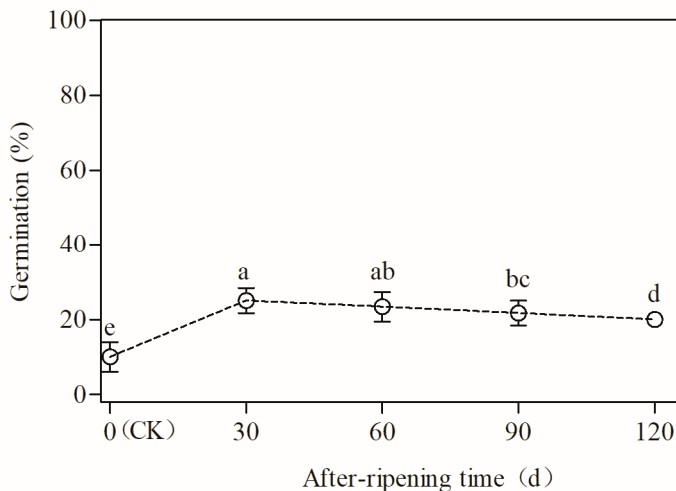

**Figure 5.** Effects of after-ripening on the dormancy release and germination of *E. ulmoides* seeds. Seeds treated with after-ripening for different days were incubated for 30 days at 10 °C in the dark and the germination percentage was calculated (mean ± standard error). Each treatment had four replicates and 30 seeds were used for each replicate. There are no significant differences between the data labeled with the same lower case letters ($p = 0.05$).

### 3.7. Effect of GA₃ Treatment on the Dormancy Release and Germination

GA is the plant hormone most closely related to seed germination. As shown in Figure 6, GA₃ significantly promoted the dormancy release and germination of the intact seeds ($f = 38.23$, $p < 0.05$). With the increase of GA₃ concentration, the germination percentage increased and then decreased. The germination percentage was $48.5 \pm 0.3\%$ at 2.0 mmol/L GA₃, which was significantly higher than that under the other GA₃ concentrations (Figure 6, $p < 0.05$).

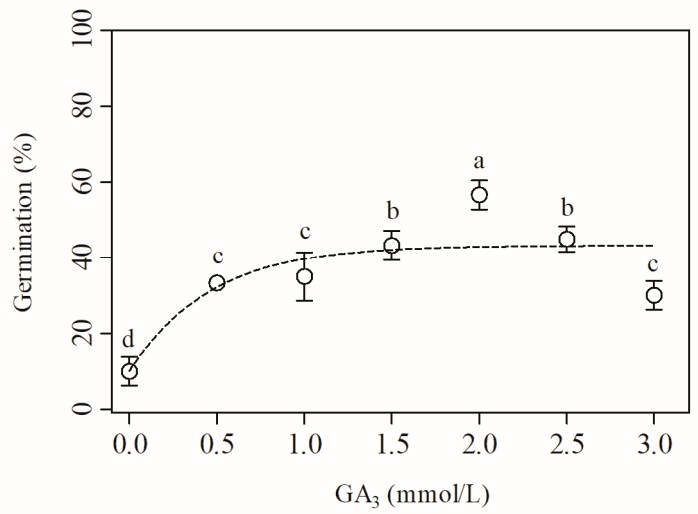

**Figure 6.** Effect of GA₃ on the dormancy release and germination of intact *E. ulmoides* seeds. The seeds were treated with different concentrations of GA₃ for 24 h, incubated in the dark at 10 °C for 30 days, and then the germination percentage was calculated (mean ± standard error). Each treatment had four replicates and 30 seeds were used for each replicate.There are no significant differences between the data labeled with the same lower case letters ($p = 0.05$).

In conclusion, a suitable temperature, a scarification treatment, a phytohormone such as GA₃, cold moisture stratification, and after-ripening were effective conditions on the dormancy release and germination of *E. ulmoides* Oliv. seeds. These results suggested that *E. ulmoides* seeds have a combinational physical and physiological dormancy.

## 4. Discussion

Seed dormancy is an environmental adaptation strategy formed during the long-term evolution of plants. Dormancy and the depth of dormancy can be determined by detecting whether fresh and mature seeds can germinate under optimal germination conditions within a specified time (usually 30 days) and germination percentages [5]. *E. ulmoides* is a monophyletic plant, which is mainly based on seed reproduction. Therefore, the study of seed dormancy has both an evolutionary and a productive significance.

### 4.1. Determination of the Dormancy Type

In this study, 13.3% of freshly harvested *E. ulmoides* seeds germinated only at 10 °C and a light/dark cycle (Figure 1). A viability measurement by TTC suggested the ungerminated seeds were not dead, indicating that *E. ulmoides* seeds had dormancy characteristics. Whether a fully developed isolated embryo can produce normal seedlings is a simple and intuitive method to determine whether the embryo has dormancy (PD) [2]. In this study, the isolated embryos could germinate and produce normal seedlings under all other temperatures and dark conditions except at 5, 30, and 0/5°C (Figure 2). These results indicated that the dormancy of *E. ulmoides* seeds might be caused by the seed coat. The probable reasons include coat-limited water absorption (physical dormancy) or the mechanical restraint of the embryo germination by the coat, i.e., a low growth potential of the embryos (physiological dormancy).

Coat scarification is one of the common methods to break the dormancy of seeds. In this study, the germination percentages of the half seeds and scarified seeds were significantly higher than those of the intact seeds (Figure 3), suggesting that *E. ulmoides* seeds likely have both PY and a shallow PD. These results were consistent with those of Yang et al. [13,34]. According to the imbibition curve, from four hours of imbibition on, the water absorption of both the scarified seeds and the isolated embryos was significantly higher than the intact seeds (Figure 4). This indicated that the seed coat significantly limited the water absorption by the embryos (Figure 4, Table 1), confirming the presence of PY in the *E. ulmoides* seeds. The seed coat of *E. ulmoides* (i.e., the peel) is rich in gutta-percha; up to 10%. Gutta-percha is insoluble in water and has a high elasticity and certain mechanical strength. This may be the reason why the seed coat is waterproof and acts as a mechanical barrier to embryo germination. Based on the above results and the dormancy classification system of Baskin and Baskin [2], we suggest that *E. ulmoides* seeds have a combinational dormancy, i.e., a physical and non-deep physiological dormancy, which is caused by a coat-limited embryo water absorption and a low embryo growth potential, respectively.

### 4.2. Effective Methods for Releasing Dormancy

Stratification is a widely used and effective method to break seed dormancy. It involves temperature, imbibition, and gravity effects [13]. In this study, cold/warm-moist stratification (Table 2) significantly promoted the seed germination. However, cold-wet stratification at 5 °C was much better than warm-wet stratification at 15 °C (Table 2). Additionally, with cold-moist stratification, the temperature window of the seed germination expanded from 10 °C with a prolonged stratification duration (Table 2). According to the dormancy classification system of Baskin and Baskin [2], the subtype of PD is likely to be a Type 3 shallow PD in *E. ulmoides* seeds.

After-ripening can remove seed dormancy in many species and the time required is closely related to the type and degree of seed dormancy [3]. Baskin and Baskin also regarded the existence of after-ripening in the dry storage process as a judgment indicator of PD [2]. In this study, the germination percentage increased by 89.5% after dry storage at room temperature for 30 days (Figure 5), indicating that this method could partially release the dormancy of *E. ulmoides* seeds. There is considerable evidence that GA enhances the growth potential of embryos by weakening the surrounding tissues, promoting the release of dormancy and the germination of seeds. In this study, soaking the seeds in 2.0 mmol/L

GA$_3$ significantly increased the germination percentage of *E. ulmoides* seeds up to 48.5% (Figure 6). This was consistent with the results of Li et al. [35,36].

Based on all the tested methods, we propose an optimal protocol for the dormancy release and germination of *E. ulmoides* seeds. Freshly collected *E. ulmoides* seeds should be selected by flotation and dried in the shade, followed by dry storage at room temperature for 30 days. This should then be followed by outdoor cold-moist stratification at 5–10 °C for 90 days in winter, resulting in the seeds being ready for sowing in the early spring of the next year.

*4.3. Seed Germination and Phylogeny of E. ulmoides*

As shown in the phylogenetic tree of seed dormancy, physical dormancy and combinational dormancy have the narrowest distributions with combinational dormancy mainly found in Leguminosae, Geraniaceae, and Cucurbitaceae [2,5,37,38]. Fossil evidence from Cercis [39] and Tilia [40] suggests that PY and PD originated in the middle and late Eocene and PD may have occurred after PY. During the Eocene–Oligocene transition [41], the climate became significant colder. That probably caused the acquisition of PD, enabling seed germination after the end of a cold winter. *E. ulmoides* also has combinational dormancy and, as the sole species in the family Eucommiaceae, may be of particular importance in the phylogenetic and evolutionary studies of plant seed dormancy and germination.

## 5. Conclusions

Fresh and mature *E. ulmoides* seeds have both physical dormancy and Type 3 shallow physiological dormancy, namely, combinational dormancy. Cold-wet stratification for 90 days effectively released the dormancy and the germination percentage reached more than 90% at 10 °C. As an ancient relic species and a monotypic taxon, this study on the dormancy characteristics is of great significance for the systematic evolution of seed dormancy and germination.

**Author Contributions:** X.W. and H.L. conceived and designed the study; Z.D. and S.D. collected the samples and performed the experiments. S.D. analyzed the data and drafted the manuscript. H.X. provided financial support, supervised the study, and revised the first draft of the manuscript. All authors have read and agreed to the published version of the manuscript.

**Funding:** This work was supported by the National Natural Sciences Foundation of China (31971646 and 31860073) and Germplasm Engineering of Characteristic Plant Resources in Enshi Prefectur (2019–2021).

**Data Availability Statement:** Not applicable.

**Acknowledgments:** We are grateful to Songquan Song (Institute of Botany, CAS) for help during the experiments and for his continuous support throughout this project.

**Conflicts of Interest:** The authors declare no conflict of interest.

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
