# Peer review of "Effects of Temperature, Scarification, Stratification, Phytohormones, and After-Ripening on the Dormancy and Germination of Eucommia ulmoides Oliv. Seeds"

_forests, doi:10.3390/f12111593_

Round 1
Reviewer 1 Report
This paper deals with the dormancy of Eucommia ulmoides Oliv. seeds and how to break it : because of the wide use of this species in China medicine and other applications, there is a need for efficient germination protocol that enable the simultaneous and robust growth of seedlings to produce efficient adult trees. The authors, aware of the dormancy classification of Baskin and Baskin, tried to get clues on the origins of the deep dormancy of these seeds, by using temperature, scarification, GA3 addition, stratification and after-ripening. They used the right experiment to ensure the ungermed seeds were not dead (viability measurement by TTC), they excised embryos and measured their dormancy, realizing that part of the dormancy was due to the seed coat. They measured water uptake kinetics.
They demonstrated with the right experiments that temperature, scarification, GA3 addition, stratification and even after-ripening influenced the germination. Consequently they determined a simple and practical protocol to break dormancy and induce germination in these seeds, by using after-ripening and stratification.
They hence made a very solid study of dormancy of these seeds and how to break it, practically but also fundamentally, determining that the dormancy of E. ulmoides was mainly due to the seed coat, and physiological plus physical.
Reviewer 2 Report
Forests Manuscript ID: forests-1446816
Titled: "Effects of Temperature, Scarification, Phytohormone, Stratification, and After-Ripening on the Dormancy and Germination of Eucommia ulmoides Oliv. Seeds"
General comments:
-In general, this manuscript has a valuable topic. The topic is scientifically sound.
- The manuscript is well written except for minor English language check required.
-The experimental design is adequately discussed.
- My main concern was the introduction and the discussion section.
-There are some minor comments.
Detailed comments:
Abstracts:
This section is missing the direct aim of the study, please write the aim as following; The aim of this study was….
Keywords:
Please change seed to be Oliv seeds
Introduction:
This section doesn’t provide enough background about the topic. I see that the introduction is very poor. This section needs to be enriched and provided with more background about the topic.
Materials and Methods:
This section is ok and the methodology is adequately described.
Results:
The results were well presented but the results were poorly discussed.
Discussion:
This section is very confusing and inadequate discussion and poor argument citations.
Please make subheadings in this section in organized way, the same order as the subheadings in the results section.
-Please rewrite this section in more organized way and relate to the data in tables and figures carefully with a comparison to the previous studies.
If it is easier to discuss the results clearly and thoroughly, the author is advised to combine the results and the discussion in one section for the best presentation and discussion to the results.
Conclusion:
Although this section not a required, It is very useful to conclude the study and for better understanding.
References:
The authors did NOT provide enough citations, and it needs to be UpToDate. (Please provide more citations from the last 5 years research.
*I am convinced that this manuscript is very valuable and will be suitable to be published in forests journal after minor revision.
